# Gut Microbiota-Derived TMAO: A Causal Factor Promoting Atherosclerotic Cardiovascular Disease?

**DOI:** 10.3390/ijms24031940

**Published:** 2023-01-18

**Authors:** Marina Canyelles, Carla Borràs, Noemí Rotllan, Mireia Tondo, Joan Carles Escolà-Gil, Francisco Blanco-Vaca

**Affiliations:** 1Institut de Recerca de l’Hospital Santa Creu i Sant Pau, Institut d’Investigacions Biomèdiques IIB Sant Pau, 08041 Barcelona, Spain; 2CIBER de Diabetes y Enfermedades Metabólicas Asociadas (CIBERDEM), 28029 Madrid, Spain; 3Department of Biochemistry and Molecular Biology, Universitat Autònoma de Barcelona, 08041 Barcelona, Spain; 4Department of Clinical Biochemistry, Hospital de la Santa Creu i Sant Pau, IIB Sant Pau, 08041 Barcelona, Spain

**Keywords:** atherosclerosis, TMAO, mortality, CVD, renal function, prospective

## Abstract

Trimethylamine-N-oxide (TMAO) is the main diet-induced metabolite produced by the gut microbiota, and it is mainly eliminated through renal excretion. TMAO has been correlated with an increased risk of atherosclerotic cardiovascular disease (ASCVD) and related complications, such as cardiovascular mortality or major adverse cardiovascular events (MACE). Meta-analyses have postulated that high circulating TMAO levels are associated with an increased risk of cardiovascular events and all-cause mortality, but the link between TMAO and CVD remains not fully consistent. The results of prospective studies vary depending on the target population and the outcome studied, and the adjustment for renal function tends to decrease or reverse the significant association between TMAO and the outcome studied, strongly suggesting that the association is substantially mediated by renal function. Importantly, one Mendelian randomization study did not find a significant association between genetically predicted higher TMAO levels and cardiometabolic disease, but another found a positive causal relationship between TMAO levels and systolic blood pressure, which—at least in part—could explain the link with renal function. The mechanisms by which TMAO can increase this risk are not clearly elucidated, but current evidence indicates that TMAO induces cholesterol metabolism alterations, inflammation, endothelial dysfunction, and platelet activation. Overall, there is no fully conclusive evidence that TMAO is a causal factor of ASCVD, and, especially, whether TMAO induces or just is a marker of hypertension and renal dysfunction requires further study.

## 1. Introduction

Atherosclerosis is a condition characterized by plaque build-up within the arteries and the thickening of the arterial walls. It is widely regarded as the leading cause of cardiovascular disease (CVD), which is currently the main cause of death in the world [1]. In recent years, the concept of pathological variation in the gut microbiota as a cause of several cardiometabolic diseases has emerged due to studies performed in both rodents and humans. Trimethylamine-N-oxide (TMAO), the main diet-induced metabolite produced by the gut microbiota, has been correlated with an increased risk of atherosclerosis, thrombosis, stroke, heart disease, type 2 diabetes, and obesity [2].

TMAO is the product of trimethylamine (TMA) oxidation in the liver. TMA can be generated by the action of the gut microbiota using dietary precursors with betaine, L-carnitine, or choline as the main sources [3]. The first publication linking TMAO and CVD was an untargeted metabolomic study published in 2011. In this study, following the adjustment for traditional cardiac risk factors and medication usage, three gut-related metabolites (TMAO, choline, and betaine) were significantly correlated with CVD [4]. Since then, the number of studies aiming to link TMAO with CVD has increased exponentially. A recent systematic review and meta-analysis of prospective studies concluded that elevated TMAO concentrations were associated with a 62% increased risk of all-cause mortality [5] as well as a 23% higher risk of cardiovascular events and a 55% higher risk of all-cause mortality [6]. However, despite these assumptions, the link between TMAO and CVD remained limited and not all the studies found a significant association [6]. Indeed, the mechanisms by which TMAO can cause an increased risk of CVD are not clearly elucidated and encompass a diverse range of pathological processes. The studied mechanisms range from cholesterol metabolism disruption, affecting atherosclerotic plaque formation, to platelet activation, leading to enhanced thrombosis activity, inflammation, and endothelial dysfunction [4,7,8].

In the present narrative review, we aimed to describe and update the causal relationship between TMAO and atherosclerotic cardiovascular disease (ASCVD), analyzing the results of three different approaches: prospective, genetic, and mechanistic studies. A literature review was performed looking for relevant published studies available on the 15 November 2022 using three electronic databases: PubMed, Google Scholar, and Web of Science. For selecting the prospective studies, we used the following combination of terms: “Cardiovascular Disease”, “Mortality”, “Cardiovascular Mortality”, or “MACE” with “TMAO” or “Trimethylamine-N-oxide” and “Prospective” or “Genetic.” The literature review was performed based on the Preferred Reporting Items for Systematic Reviews and Meta-Analyses statement and the process was agreed upon by all authors.

## 2. Trimethylamine-N-Oxide (TMAO)

### 2.1. Microbial Pathways Leading to TMA Production in the Gut

Various gut microorganisms can form TMA from dietary precursors. In the gut microbiota, several families of bacteria, such as Firmicutes, Proteobacteria, and Actinobacteria phyla, isolated from commensal bacteria in the human intestine, have been identified as choline and carnitine consumers and, therefore, potential TMA producers. Bacteroidetes, however, appear to be unable to produce TMA [9]. The choline utilization gene cluster, which is responsible for anaerobic choline degradation, has been shown to contain a gene coding for the specific glycyl radical enzyme choline TMA-lyase and its corresponding radical S-adenosyl-L-methionine activating protein, which uses choline as a substrate [10]. Another microbial metabolic pathway via which TMA is generated is a two-component oxygenase/reductase system, which uses carnitine as a substrate [11] Lastly, another bacterial lyase is responsible for TMA production, using choline, betaine, L-carnitine, and γ-butyrobetaine (γBB) as a substrate [12]. 

Inter-individual variability in the composition of the gut microbiota can be influenced by dietary patterns. A greater increase in post-prandial plasma TMAO was observed following consuming both eggs and beef compared to fruit, which was used as a control. High TMAO producers had a higher number of Firmicutes than Bacteroidetes and a less diverse gut microbiome. More specifically, *Clostridiaceae*, *Lachnospiraceae* and *Veillonellaceae*, Clostridiales of the Firmicutes phylum were the most abundant genre. Conversely, low TMAO producers had a greater number of Bacteroidales of the Bacteroidetes phylum, of which *Bacteroidaceae* and *Prevotellaceae* were predominant [13].

### 2.2. Dietary Precursors of TMA and TMAO

TMAO can be ingested directly from food. It is found in fish and marine invertebrates such as mussels or other seafood. Concentrations in fresh seafood can vary considerably between species, habitat, depth, and season. Compared to other dietary sources, fish has the biggest impact on TMAO concentrations. The consumption of fish yielded 50 times higher postprandial circulating TMAO concentrations than the consumption of other foods rich in choline or carnitine [14]. Furthermore, TMAO can also be produced by the oxidation of TMA generated by the gut using different dietary precursors: L-carnitine, choline, N6,N6,N6-trimethyl-L-lysine (trimethyllysine, TML), betaine, and ergothioneine. A schematic presentation of the major microbial pathways leading to TMA/TMAO synthesis from dietary precursors is shown in Figure 1.

L-carnitine is synthesized endogenously from methionine and lysine in the liver, kidneys, heart, testes, and brain. In certain conditions, an exogenous source may be needed to supplement endogenous synthesis. Foods of animal origin (especially meat and dairy products) are rich in L-carnitine. It can also be found in grains and vegetables but in lower concentrations [15]. Choline can also be considered a conditionally essential nutrient. As the endogenous production by the liver may not be enough for human requirements, it needs to be obtained from the diet. Choline is present in foods in different forms, mainly in water-soluble forms that include free choline, phosphocholine, and glycerophosphocholine, and lipid-soluble forms that include phosphatidylcholine and sphingomyelin. The main sources of choline are eggs and liver, followed by fish, whole-grain cereals, vegetables, fruit, milk, fat, and oils [16]. TML is a non-protein amino acid with an important role as a precursor of L-carnitine. TML is present as a free amino acid in legumes, grains, leafy and Solanaceae vegetables, and fruit [17].

### 2.3. TMAO Metabolism

Once the gut microbiota has produced TMA from dietary precursors, it crosses the intestinal lumen. TMA can only be produced and absorbed by the small intestine, and it is then transported via portal circulation to the liver [18]. TMA is afterward converted into TMAO by flavin monooxygenase (FMO) in the liver of the host. Among the five members of the FMO family, only FMO1 and FMO3 have the ability to oxidize TMA to TMAO. FMO3 is the main isoform expressed in the human liver (Figure 1) [19]. 

Interestingly, a sexually dimorphic expression pattern of FMO3 has been observed in both mice and humans, with females showing higher expression than males [20]. This gender difference may be explained by hormonal regulation, as testosterone is responsible for the lower hepatic FMO3 expression in males and estrogen induces FMO3 gene expression in females [21]. Studies on humans have produced divergent results on gender-related differences in plasma TMAO concentrations. Several studies investigating TMAO expression in humans did not find significant differences between the sexes in terms of plasma TMAO levels [22,23,24]; others, however, reported higher levels either in females [19,25] or in males [26,27]. 

Urinary excretion is the major elimination pathway of TMAO from systemic circulation. TMAO is almost exclusively eliminated by the kidneys in an unmodified form via filtration and active secretion mechanisms (Figure 1). A kinetic study in 40 healthy men with d9-TMAO revealed that TMAO can be detected in plasma only 15 min after oral ingestion. The elimination of TMAO was greatest during the first six hours (showing an enrichment of 67%) after consumption of the tracer. An estimated 96% of the administrated d9-TMAO was excreted unmodified in the urine after 24 h, leaving only 4% retained in the body, with none detected in the feces [28].

Active secretion mechanisms, which are implicated in TMAO excretion, occur through different transporters. The most important transporters are the organic cation transporters (OCT). Specifically, OCT1 to 3 are TMAO transporters, and OCT1 and OCT2 are decisive in TMAO kinetics in mice. TMAO can be absorbed by OCT2 in humans, but its contribution to tubular secretion is not clearly observed under normal conditions [29]. OCT1/2 knockout mice present higher levels of TMAO than wild-type mice. OCT1 is mainly located in the liver and OCT2 is in the kidney. The increase in TMAO levels is a consequence of low renal excretion; the liver/plasma ratio of TMAO is not altered, but the kidney/plasma ratio of TMAO is decreased. Moreover, OCT2 is physiologically relevant to TMAO renal tubular uptake across all the concentrations observed in humans [30].

## 3. TMAO in Prospective Studies

After Wang et al. linked TMAO with CVD in 2011, several studies have been carried out, with somewhat controversial results. While some reported that TMAO independently predicted mortality or major adverse cardiovascular events (MACE), others did not find independent associations between TMAO levels and cardiovascular outcomes. In this review, we focused on prospective studies, since they are more powerful in assessing the risk association between an exposition or molecule and an outcome. The first meta-analysis of published prospective studies concluded that higher circulating levels of gut-related metabolites, including TMAO, were associated with an increased risk of MACE, regardless of conventional risk factors [5]. However, only one year later, another meta-analysis including 11 prospective studies found that, despite high circulating TMAO being associated with an increased risk of cardiovascular events and all-cause mortality, the link between TMAO and CVD remained limited and inconsistent [6]. Moreover, the most recent systematic review, which examined the relationship of TMAO with MACE, cardiovascular mortality, and all-cause mortality, also found that TMAO was positively associated with these outcomes in some but not all of the included studies [31]. There are several limitations to be considered in systematic reviews and meta-analyses. One of them is the high heterogeneity of the studies, which, in some cases, may not equally take into account the type of cohort included (general population, pre-existing, cardiovascular risks, etc.). Another is the different definition of elevated TMAO levels or confounding factors, such as environment (diet or lifestyle), not being taken into account.

Table 1 includes larger published prospective studies evaluating the association of TMAO and adverse outcomes (cardiovascular-related and mortality-related) in CVD-free individuals, patients with CVD or coronary artery disease (CAD), patients with acute coronary syndrome (ACS), and patients with chronic kidney disease (CKD). The table includes analyses of individuals free of CVD, as it is a population unbiased by any specific disease or condition; the CAD and ACS patients are a representation of systemic CVD; and the CKD patients were included because TMAO is almost exclusively cleared via kidney and thus, renal function is the most well-known confounder. Only studies with clear results in terms of hazard ratio, confidence interval, and *p* values were included in the table. Depending on the studied population, the outcome evaluated, or the covariates included in the adjustment, different results were found.

In community-based studies, adjustment for estimated glomerular filtration rate (eGFR) [32] and age and sex [33] showed a non-significant association between TMAO and all-cause mortality. Moreover, no significant association was found between TMAO and ASCVD [34]. In patients with prevalent CVD or CAD the most significant association was found with all-cause mortality [35,36] rather than MACE [26] or ASCVD [34,35]. In patients with ACS, a significant association with major outcomes was also found, but renal function was not included in the adjustment analysis [37]. Only short-term outcomes, with a follow-up of one or two months, maintained a significant association after adjustment for eGFR [38]. Last, the CKD patient results also showed a significant association when eGFR was not included in the adjustment [39], while adjustment for renal function led to divergent results [40,41]. 

Renal function, mainly calculated as the eGFR, is the main covariate used in adjustment. Correction for renal function almost always translated into a decrease or loss in the significant association between TMAO and the outcome—independent of the target population. Considering this observation, it could be argued that TMAO’s association with the outcomes may be mediated—at least partially—by renal function [42]. Moreover, TMAO does not seem to add an additional risk to that already provided by impaired renal function or CKD alone. TMAO could be just a marker of renal impairment or directly contribute to decreased renal function, as it was observed in mice [43]. This rather suggests that elevated circulating concentrations of TMAO may have a direct effect on renal impairment, which, in turn, is a determinant of CVD risk [44]. An important limitation of these studies is that the use of antibiotics and modifications in dietary habits were not always recorded as confounding factors. 

**Table 1 ijms-24-01940-t001:** Prospective studies about TMAO and adverse outcomes.

Year	N	Follow-Up	Outcomes	Covariates in the Adjusted Models	Hazard Ratio (95% CI) and *p* Value	Ref.
**Individuals free of known CVD: community-based population**						
2013	4007	3 years	MACE	1: age, sex, smoking status, SBP, LDLc, HDLc, diabetes, CRP 2: model 1 + myeloperoxidase level, eGFR, total white-cell count, BMI, medication 3: model 2 + extent of the disease	UA: 2.54 (1.96–3.28) *p* < 0.001 1: 1.88 (1.44–2.44) *p* < 0.001 2: 1.49 (1.10–2.03) *p* = 0.01 3: 1.43 (1.05–1.94) *p* = 0.02	[45]
2021	4131	15 years	ASCVD	Age, sex, race, study site, education, income, health status, smoking status, alcohol intake, physical activity, BMI, WC, lipid-lowering medication, antihypertensive medication, antibiotics, T2D, HDL-C, LDL-C, TG, CRP, SBP, DBP, diet, eGFR	Adj: 1.07 (0.90–1.27) *p* = 0.516 Adj: 1.37 (0.98–1.90) *p* = 0.084 (individuals with eGFR < 60 mL/min 1.73 m^2^) Adj: 0.98 (0.80–1.20) *p* = 0.903 (individuals with eGFR ≥ 60 mL/min 1.73 m^2^)	[34]
2022	5333	13.2 years	All-cause mortality Cardiovascular mortality Non-cardiovascular mortality	1: age, sex, race, and enrollment site 2: model 1 + education, household income, smoking, BMI, physical activity, treated hypertension, instrumental activities of daily living, self-reported health status, systolic blood pressure, HDL cholesterol, prevalent atrial fibrillation, prevalent coronary heart disease, myocardial infarction, prevalent diabetes, prevalent chronic obstructive pulmonary disease, and reported daily intake of eggs, fish, liver, non-processed red meat, processed meat, and total calories 3: model 2 + eGFR	1: 1.36 (1.24–1.51) *p* < 0.001 2: 1.30 (1.17–1.44) *p* < 0.001 3: 1.07 (0.96–1.19) *p* = 0.51 1: 1.50 (1.27–1.77) *p* < 0.0001 2: 1.35 (1.14–1.60) *p* < 0.0001 3: 1.09 (0.91–1.30) *p* = 0.65 1: 1.29 (1.14–1.46) *p* < 0.0001 2: 1.27 (1.12–1.44) *p* = 0.0003 3: 1.06 (0.93–1.21) *p* = 0.61	[32]
2022	6393	10.5 years	All-cause mortality Cardiovascular mortality Non-cardiovascular mortality	1: age and sex 2: model 1 + HT, diabetes mellitus, total cholesterol, smoking, BMI, and eGFR	UA: 3.07 (2.48–3.80) *p* < 0.001 1: 1.09 (0.88–1.37) *p* = 0.43 2: 1.11 (0.88–1.40) *p* = 0.38 UA: 3.75 (2.56–5.49) *p* < 0.001 1: 1.13 (0.77–1.70) *p* = 0.54 2: 0.97 (0.65–1.46) *p* = 0.88 UA: 2.77 (2.14–3.60) *p* < 0.001 1: 1.07 (0.82–1.40) *p* = 0.64 2: 1.17 (0.88–1.54) *p* = 0.28	[33]
**Patients with CVD or CAD**						
2015	339	8 years	MACE	eGFR	UA: 1.01 (0.98–1.04) *p* = 0.493 Adj: 0.991 (0.98–1.00) *p* = 0.062	[26]
2016	2235	5 years	All-cause mortality	1: age, sex, SBP, LDL-C, HDL-C, smoking, and diabetes mellitus 2: model 1 + hsCRP and eGFR 3: model 2 + medications (angiotensin-converting enzyme inhibitor, angiotensin-receptor blocker, β-blocker, aspirin, or statin), number of stenotic vessels, myeloperoxidase, and BNP	UA: 3.90 (2.78–5.48) *p* < 0.0001 1: 2.61 (1.82–3.76) *p* < 0.0001 2: 1.95 (1.33–2.86) *p* = 0.003 3: 1.71 (1.11–2.61) *p* = 0.0138	[36]
2021	1287	15 years	ASCVD	Age, sex, race, study site, education, income, health status, smoking status, alcohol intake, physical activity, BMI, WC, lipid-lowering medication, antihypertensive medication, antibiotics, T2D, HDL-C, LDL-C, TG, CRP, SBP, DBP, diet, eGFR	1.10 (0.87–1.39) *p* = 0.179	[34]
2021	449	9 years	All-cause mortality	1: BMI, diabetes status, eGFR 2: model 1 + age, sex, smoking status, SBP, DBP, hsCRP, HDL-C, LDL-C, white blood cell count, and in LIFE-CAD additionally for presence of CAD	UA: 1.61 (1.36–1.92) *p* < 0.001 1: 1.30 (1.07–1.58) *p* = 0.009 2: 1.24 (1.01–1.51) *p* = 0.040	[35]
2022	1726	5 years	All-cause mortality Cardiovascular mortality AMI	1: age, gender, and history of CAD 2: pre-defined patient characteristics, cardiovascular risk factors, and medical history, including age, gender, body mass index, smoking history, positive cardiovascular family history, hypertension, hypercholesterolemia, history of diabetes, history of stroke, history of CAD, previous AMI, history of heart failure and adjudicated functionally relevant coronary artery disease* Both models also adjusted adding Cystatin-C	UA: 1.42 (1.29–1.58) *p* < 0.001 1: 1.28 (1.13–1.43) *p* < 0.001 1 *: 1.16 (1.03–1.30) *p* = 0.013 2: 1.23 (1.09–1.38) *p* < 0.001 2 *: 1.11 (0.99–1.26) *p* = 0.084 UA: 1.60 (1.39–1.84) *p* < 0.001 1: 1.44 (1.4–1.68) *p* < 0.001 1 *: 1.28 (1.10–1.48) *p* < 0.001 2: 1.36 (1.16–1.58) *p* < 0.001 2 *: 1.19 (1.01–1.40) *p* = 0.032 UA: 1.38 (1.17–1.65) *p* < 0.001 1: 1.33 (1.11–1.60) *p* = 0.002 1 *: 1.22 (1.01–1.48) *p* = 0.043 2: 1.32 (1.08–1.60) *p* = 0.006 2 *: 1.17 (0.95–1.45) *p* = 0.146	[46]
2022	4132	9.8 years	All-cause mortality Cardiovascular mortality Non-cardiovascular mortality	1: age and sex 2: model 1 + HT, diabetes mellitus, total cholesterol, smoking, BMI, and eGFR	UA: 2.29 (1.88–2.79) *p* < 0.001 1: 1.32 (1.08–1.62) *p* = 0.01 2: 1.07 (0.86–1.32) *p* = 0.56 UA: 2.73 (2.00–3.72) *p* < 0.001 1: 1.52 (1.11–2.09) *p* = 0.01 2: 1.16 (0.83–1.62) *p* = 0.38 UA: 2.02 (1.57–2.61) *p* < 0.001 1: 1.20 (0.92–1.56) *p* = 0.18 2: 1.02 (0.77–1.34) *p* = 0.90	[33]
**ACS patients**						
2017	1079	2 years	All-cause mortality Death/MI	Age, sex, SBP, heart rate, past histories of MI/angina, increased BP and diabetes, Killip score, STEMI class, revascularization, and medication	UA: 1.21 (0.98–1.48) *p* = 0.074 UA: 1.4 (1.26–1.55) *p* < 0.0005 Adj: 1.21 (1.03–1.439) *p* = 0.023	[37]
2017	530	30 days 6 months 7 years	MACE (30 days and 6 months) All-cause mortality (7 years)	Age, gender, HDL-C, LDL-C, smoking, presence or absence of a history of diabetes mellitus, HT, hyperlipidemia, revascularization or CAD, CRP level, eGFR, initial cTnT level, diagnosis of either STEMI, nonSTEMI, or unstable angina	UA: 2.29 (1.39–3.79) *p* < 0.01 Adj: 6.3 (1.89–21.00) *p* < 0.01 UA: 2.48 (1.51–4.09) *p* < 0.001 Adj: 5.65 (1.91–16.74) *p* < 0.01 UA: 3.72 (2.23–6.20) *p* < 0.001 Adj: 1.81 (1.04–3.15) *p* < 0.05	[38]
2021	326	9 years	All-cause mortality	1: BMI, diabetes status, eGFR 2: model 1 + age, sex, smoking status, SBP, DBP, hsCRP, HDL-C, LDL-C, white blood cell count, and in LIFE-CAD additionally for presence of CAD	UA: 1.39 (0.98–1.97) *p* = 0.066 1: 1.10 (0.74–1.65) *p* = 0.638 2: 1.07 (0.70–1.63) *p* = 0.771	[35]
2021	292	7 years	Cardiovascular mortality	Prior MI, diabetes, age, gender, diabetes, hemoglobin and creatinine levels at admission	UA: 21.33 (4.7–96.5) *p* < 0.001 Adj: 11.62 (2.3–59.7) *p* = 0.003	[47]
2022	309	6.7 years	MACE	1: age, sex, BMI, smoking, HT, dyslipidemia, type 2 diabetes, unstable angina, STEMI, nonSTEMI, statin medication, beta-blockers, oral antidiabetic medication, insulin medication, diuretics, aspirin 2: model 1 + eGFR	UA: 2.63 (1.69–4.08) *p* < 0.001 1: 1.83 (1.08–3.09) *p* = 0.043 2: 1.66 (0.98–2.82) *p* = 0.119	[42]
**Stable CKD patients**						
2015	521	5 years	All-cause mortality	Age, sex, systolic blood pressure, LDL-C, HDL-C, smoking, diabetes mellitus, hsCRP, and eGFR	UA: 2.76 (1.74–4.37) *p* < 0.001 Adj: 1.93 (1.13–3.29) *p* < 0.05	[41]
2016	2529	3 years	Ischemic events	1: age, sex, race, diabetes and cardiovascular comorbidities at baseline 2: model 1 + traditional cardiovascular risk factors 3: model 2 + chronic kidney disease-specific risk factors 4: Adjusted for all covariates with statically significant association with ischemic events	UA: 1.45 (1.28–1.64) 1: 1.38 (1.21–1.57) *p* < 0.0001 2: 1.39 (1.22–1.59) *p* = 0.0055 3: 1.24 (1.07–1.43) *p* = 0.065 4: 1.23 (1.06–1.42) *p* = 0.0059	[39]
2016	339	3.3 years	All-cause mortality	1: age, race, sex 2: model 1 + SBP, LDL-C, HDL-C, smoking, CRP, eGFR	1: 2.45 (1.09–5.50) *p* = 0.02 2: 1.25 (0.48–3.28) *p* = 0.37	[40]

Adj: adjusted; AMI: acute myocardial infarction; ACS: acute coronary syndrome; BNP: brain natriuretic peptide; BMI: body mass index; CAD: coronary artery disease; CKD: chronic kidney disease; cTnT: cardiac troponin T; CVD: cardiovascular disease; DBP: diastolic blood pressure; eGFR: estimated glomerular filtration rate; HDL-C: high-density lipoprotein cholesterol; HR: hazard ratio; hsCRP: high sensitivity C reactive protein; HT: hypertension; LDL-C: low-density lipoprotein cholesterol; MACE: major adverse cardiovascular event; SBP: systolic blood pressure; STEMI: ST-segment elevation myocardial infarction; TG: triglycerides; UA: unadjusted; WC: waist circumference.

Other target populations were also analyzed in other prospective studies, such as heart failure, type 2 diabetes (T2D), peripheral artery disease, and hemodialysis treatment, among others. The results of these studies are summarized in Appendix A. As previously noted, the association tends to remain only significant when renal function is not included in the adjustment analysis.

Focusing on the relationship of TMAO with atherosclerotic burden, there are also controversial results. The TMAO levels were independently associated with high coronary atherosclerotic burden in NSTEMI [43] and STEMI [44] patients. A specific gut microbiota profile was found to be associated with coronary artery calcium (CAC) in patients without previously reported CVD [45]. However, TMAO was not associated with different measures of atherosclerosis, including CAC, in another population-based study [46].

## 4. TMAO in Genetic Studies

The results of observational studies are not always consistent, and, therefore, it is at times difficult to establish the causality relation between TMAO and CVD. Mendelian randomization (MR) uses measured variation in genes of known function, in this case influencing TMAO plasma concentration on a disease or on several diseases. An MR method is somewhat similar to a randomized controlled trial, where genetic alleles are randomly assorted during conception, and is thus less likely to be affected by confounding or reverse causation. For this reason, it has been widely accepted as a method to explore the potential causal effect of exposure on disease [48]. Only three MR studies were performed focusing on the role of TMAO. The first one assessed the association of gut-microbiota-related metabolites and cardiometabolic health [49]; the second was in relation to Alzheimer’s disease [50]; and the most recent was in relation to blood pressure [51]. In the study of Jia et al., they did not find a significant association of genetically predicted higher TMAO levels with cardiometabolic disease (including T2D, atrial fibrillation, CAD, MI, stroke, and CKD). They also found that T2D and CKD were causally associated with an increase in TMAO levels, suggesting that the association of TMAO and CVD may be due to confounding or reverse causality [49]. On the other hand, Wang et al. found a positive causal relationship between TMAO and systolic blood pressure—but not with diastolic blood pressure [51]. One recent meta-analysis also found a significant positive dose-dependent association between circulating TMAO levels and hypertension prevalence regardless of different stratifications [52]. The specific mechanism by which TMAO could affect blood pressure is currently unknown, and all proposed are also linked to the arteriosclerosis-induced capacity of TMAO. 

## 5. Mechanisms Related to TMAO Atherogenic Effect

### 5.1. TMAO, Cholesterol Metabolism Disruption, and Atherosclerotic Plaque Formation

Experimental evidence indicates that TMAO-mediated foam cell formation in atherosclerotic plaques may occur through multiple pathways. Early evidence demonstrated that TMAO enhanced macrophage cholesterol uptake via receptor cluster of differentiation (CD) 36 and scavenger receptor A upregulation [4]. Furthermore, in another study, TMAO-mediated enhancement of CD36 expression and foam cell formation was also induced by oxidized LDL, and this was dependent on the MAPK/JNK pathway [53]. Furthermore, TMAO induces the expression of stress-inducible heat shock proteins (HSP) such as glucose-regulated protein 94 and HSP70 in mouse macrophages, leading to the induction of endoplasmic reticulum stress [54].

On the other hand, some human studies reported low HDL cholesterol levels in high TMAO-expressing individuals with CVD [36,55], thereby indicating that HDL dysfunction could be mechanistically linked to the proatherogenic effects of TMAO. However, other reports found a positive correlation or neutral relationship between HDL cholesterol and CVD [22,56]. The ability of HDL to induce macrophage cholesterol efflux, the first step of reverse cholesterol transport (RCT), is considered the main HDL atheroprotective function. Several reports have shown divergent results regarding the TMAO-mediated effects on macrophage transporters involved in cholesterol efflux, specifically indicating enhancing [57], neutral [58], or negative effects [59,60] on ATP binding cassette (ABC) A1. We recently reported that TMAO and its major gut microbial precursors did not produce significant effects on HDL-mediated macrophage cholesterol efflux [61], indicating that the first step in the RCT process was not affected by TMAO. In contrast, the suppression of the intestinal microbiota was correlated with a two-fold stimulation of the overall macrophage-to-feces RCT in mice. This change was concomitant with the enhancement of the last RCT step, the biliary secretion of bile acids [62]. In line with these findings, dietary supplementation with TMAO impaired the overall RCT transport from macrophages in mice, and antibiotic treatment reversed this TMAO-dependent downregulation of RCT. In this report, TMAO mainly affected the major pathways for bile acid synthesis and liver transport that control their elimination from the body [57]. Mechanistically, farnesoid X receptor (FXR), a nuclear receptor that acts as a sensor of intracellular bile acid levels, plays a critical role in TMAO-mediated bile acid disruption, since liver FMO3 expression is upregulated by dietary bile acids through an FXR-mediated pathway [19]. Importantly, FMO3 was also reported to be a negative regulator of transintestinal cholesterol export [63], a process responsible for 30% of the total cholesterol loss through the feces in mice [64]. Consistent with these findings, liver FMO3 knockdown also reduced liver and plasma lipid levels, the bile acid pool size, and glucose and insulin levels and prevented atherosclerosis in LDL receptor-deficient mice [65], suggesting multiple mechanistic processes involved in TMAO/FMO3-mediated foam cell formation in vivo. The main pathophysiological mechanisms involved in atherosclerotic plaque formation are shown in Figure 2.

**Figure 2 ijms-24-01940-f002:**
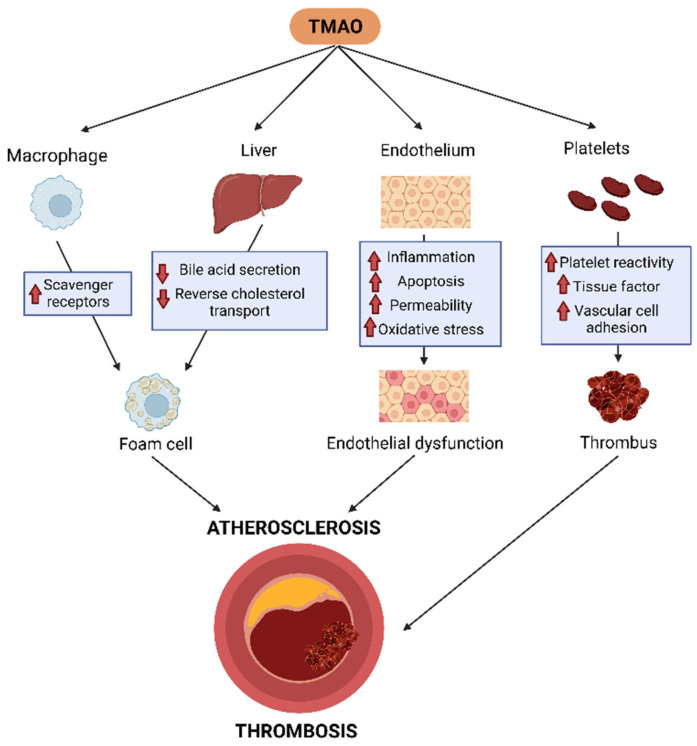
TMAO enhances proatherogenic and prothrombotic mechanisms. TMAO promotes macrophage foam cell formation, impairs liver cholesterol metabolism, affects reverse cholesterol transport, and induces endothelial dysfunction and platelet aggregation by a variety of mechanisms. TMAO: Trimethylamine-N-oxide.

### 5.2. TMAO, Inflammation, and Endothelial Dysfunction

Endothelial cell dysfunction in the vasculature is considered a key factor in the early stages of atherosclerosis. This process is characterized by higher production of reactive oxygen species (ROS) and pro-inflammatory factors but also a deficiency in nitric oxide (NO) bioavailability, which regulates the vascular tone [66]. Indeed, TMAO elevated interleukin 6, C reactive protein, tumor necrosis factor alpha, and ROS in endothelial cells, whereas it reduced NO production [67]. TMAO also induced the expression of cytokines and adhesion molecules in endothelial cells, thereby promoting monocyte adherence [68]; this induction required the nuclear factor kappa-light-chain-enhancer of activated B cells signaling pathway [8] (Figure 2).

Other lines of evidence indicate that TMAO is inversely related to artery endothelial function in vivo, as assessed in one study via brachial artery flow-mediated dilation in elderly patients. In the same work, TMAO supplementation also induced endothelial dysfunction via reduced NO bioavailability and increased superoxide-driven oxidative stress in young mice [69]. In line with these findings, 3,3-Dimethyl-1-butanol, an inhibitor of TMAO production, normalized the expression of pro-inflammatory cytokines and superoxide production, whereas it reduced endothelial nitric oxide synthase (eNOS) expression in the aorta of old rats [70].

Mechanistically, strong evidence indicates that TMAO promotes different inflammatory signaling pathways, particularly the NOD-, LRR-, and pyrin domain-containing protein 3 (NLPR3), which is also involved in enhanced oxidative stress, endothelial cell apoptosis, and hyperpermeability (Figure 2). Accordingly, TMAO significantly enhanced oxidative stress and activated the TXNIP-NLRP3 inflammasome in human umbilical vein endothelial cells, and these effects were reversed by the ROS inhibitor N-acetylcysteine [71]. These data were confirmed by another study that demonstrated TMAO induced the formation and activation of NLRP3 and enhanced IL-1β production in carotid artery endothelial cells. These effects were also blunted by ROS scavengers and the inhibition of lysosomal cysteine protease cathepsins B [72], thereby suggesting that TMAO induces the inflammasome via both lysosomal destabilization and ROS release. It has also been reported that TMAO upregulated NLRP3, pyroptosis, and succinate dehydrogenase complex subunit B (SDHB) in vascular endothelial cells of dyslipidemic apoE-deficient mice. TMAO also impaired the mitochondria’s structure and function and increased ROS generation by upregulating SDHB expression in cultured vein endothelial cells [73], thereby identifying an alternative TMAO-mediated proatherogenic mechanism.

The integrity of the endothelium is maintained by tight junction proteins. In this context, TMAO downregulated the tight junction protein-1 expression on the endothelial cells, and this was prevented by silencing NLRP3. Furthermore, the TMAO-mediated increase in endothelial cell permeability was also blunted by NLRP3 siRNA transfection [72]. On the other hand, TMAO treatment also caused endothelial cell disjunction via high-mobility group box 1, and this caused the further activation of pro-inflammatory toll-like receptor (TLR) 4. Interestingly, TLR4 inhibition attenuated TMAO-induced junction protein disruption [74].

### 5.3. TMAO, Platelet Activation, and Thrombosis

The first proposed mechanism by which TMAO enhances thrombosis was related to its capacity to directly increase platelet sensibility to multiple agonists, also increasing thrombosis in vivo and stimulating platelet hypersensitivity, enhancing the release of intracellular Ca^2+^ (Figure 2). Dietary precursors of TMAO, such as choline, also increase platelet sensitivity as well as thrombosis in vivo, with a key role played by the microbiota [7]. Moreover, oral choline supplementation causes an increase in platelet aggregation responses to the platelet agonist adenosine diphosphate (ADP) and strikes the dose-dependent association between TMAO levels and platelet function [75]. 

3,3-dimethyl-1-butanol (DMB), a drug that inhibits microbial choline TMA-lyase activity in vitro and in vivo [76], was shown to be a therapeutic strategy for reducing thrombosis potential. DMB significantly reduced both the TMAO levels and ADP-dependent platelet aggregation in platelet-rich plasma (PRP); this reduction was completely reversed by the direct injection of TMAO [77]. Indeed, germ-free mice colonized with a microbiome from a high TMAO producer donor presented a significantly increased platelet aggregation response and platelet responsiveness [78], supporting the link among the microbiome, TMAO, and thrombosis in vivo (Figure 2).

TMAO also increases thrombosis in vitro, since mouse PRP incubated with high levels of TMAO enhanced platelet aggregation through two different agonists—ADP and collagen. In addition, PRP from FMO3-knockdown mice showed a significant decrease in platelet aggregation response induced by ADP stimulation. FMO3 suppression significantly caused decreased collagen-dependent platelet adhesion to the matrix in whole blood, decreased platelet responsiveness, slower clot formation, and significantly longer times to vessel occlusion [79,80]. Furthermore, decreased expression levels of inflammatory genes were found, but no differences were observed in terms of other inflammatory or thrombotic-related biomarkers, such as E-selection, intercellular adhesion molecule 1, and Von Willebrand factor [80]. On the other hand, FMO3 overexpression in mice induced heightened platelet responsiveness to ADP, faster clot formation, and a significantly shorter time to vessel occlusion [79]. 

The most recent proposed mechanism was the ability of TMAO to induce tissular factor (TF) expression and activity in endothelial vascular cells as a cause of atherosclerotic thrombosis (Figure 2). Incubating TMAO with human coronary endothelial cells causes an increase in TF expression (mRNA, protein, and surface expression) as well as enhances TF activity and thrombin production [81]. Moreover, TMAO causes a dose-dependent overexpression of TF and vascular cell adhesion molecule type 1 in both human and mouse cells. This TMAO-dependent enhancement in TF expression was shown to cause an increase in thrombogenicity in mouse models and could be reverted through a TF-inhibitory antibody or TMA-lyase inhibitor. Overall, endothelial TF contributes to TMAO-related arterial thrombosis potential and can be specifically blocked by the targeted non-lethal inhibition of gut microbial choline TMA-lyase [82].

## 6. Conclusions

During the last decade, the number of studies focusing on the association of TMAO with CVD increased year after year. However, the results of prospective studies failed to establish a clear relationship between TMAO and CVD or related events, with a key role of renal function as a potential confounding factor. However, it is also possible that increased TMAO causes renal dysfunction—at least in part through hypertension. Experimental data have proposed several TMAO-mediated proatherogenic and prothrombotic mechanisms, including the disruption of cholesterol metabolism, inflammation, endothelial dysfunction, and platelet activation. However, the impact of these detrimental effects of TMAO by itself may be not enough to be clinically meaningful. Overall, the potential of TMAO as a causal agent of ASCVD in humans remains to be fully clarified.

## Figures and Tables

**Figure 1 ijms-24-01940-f001:**
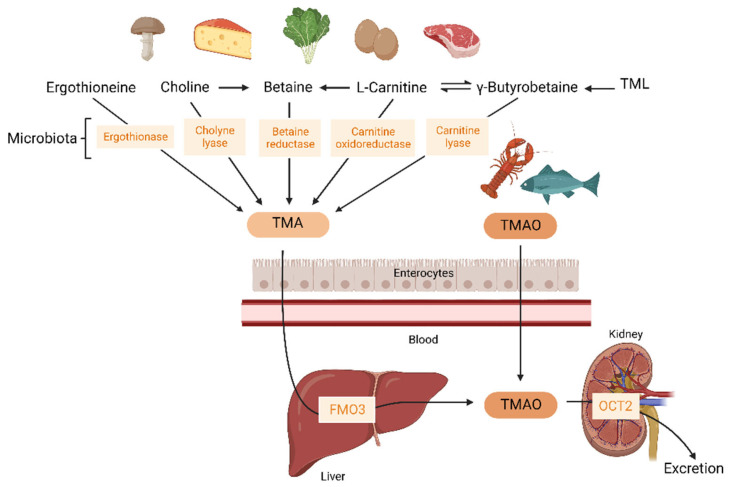
Schematic representation of TMA/TMAO formation and TMAO excretion. TMA is obtained from dietary precursors, such as ergothioneine, choline, betaine, L-carnitine, γ-butyrobetaine, and TML metabolized by microbial enzymes. TMA crosses the intestinal barrier and, in the liver, is converted into TMAO by FMO3. TMAO can be directly obtained from fish and marine invertebrates. It crosses the intestinal barrier and is mainly removed by renal excretion. TMAO could interact with endothelial cells in circulation during its metabolism, after oxidation in the liver, and before entering into the kidneys. The potential effects of circulating TMAO on atherogenic processes are shown in Figure 2. TMA: Trimethylamine-mine; TMAO: Trimethylamine-N-oxide; TML: Trimethyllysine; FMO3: flavin monooxygenase 3; OCT2: organic cation transporter 2.

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
