# Peer review of "Gut Microbiota-Derived TMAO: A Causal Factor Promoting Atherosclerotic Cardiovascular Disease?"

_ijms, 2023, doi:10.3390/ijms24031940_

Round 1

Reviewer 1 Report

Please see below the review report of paper no. ijms-2147558, Title: "Gut microbiota-derived TMAO: a causal factor promoting atherosclerotic cardiovascular disease?"

The reviewer hopes that his point of view could help the authors improve their work well. 

Your work is highly appreciated.

Sincerely yours,

The Reviewer

 Review report:

Trimethylamine-N-oxide (TMAO) is a compound produced by the gut microbiota and eliminated through the kidneys. Some studies have suggested that high levels of TMAO may increase the risk of developing atherosclerotic cardiovascular disease (ASCVD) and other cardiovascular complications, such as heart attacks and stroke. However, the relationship betwank you very much.een TMAO and ASCVD is not fully understood, and the results of different studies have been inconsistent. The association between TMAO and ASCVD may be mediated by kidney function. The mechanisms by which TMAO might increase the risk of ASCVD are not well understood. Still, it has been suggested that TMAO may alter cholesterol metabolism, cause inflammation, and disrupt the functioning of blood vessels and platelets. More research is needed to determine if TMAO is a causal factor in ASCVD or is a marker of other health conditions such as hypertension and kidney dysfunction.

Here are some comments and suggestions for the authors:

-          Line 37: "Introduction" should be "1. Introduction"

-          Line 37-line 71:

The first comment about this part is that it presents a somewhat one-sided view of the evidence linking TMAO to ASCVD, as it only mentions studies that have found a positive association between the two. It would be helpful also to consider studies that have not found a significant association between TMAO and ASCVD to provide a more balanced view of the evidence. Additionally, this paper does not discuss potential confounding factors that may influence the relationship between TMAO and ASCVD, such as other lifestyle or dietary factors that could affect TMAO levels and ASCVD risk. It would be helpful to consider these factors to understand better the nature of the relationship between TMAO and ASCVD.

-          Line 73-line 94:

The authors should provide more context and explanations for some of the technical terms and processes mentioned in the article. Additionally, the paper does not discuss the potential health consequences of differences in TMAO production or the mechanisms by which TMAO may affect health. It would be helpful to consider this information to understand the findings' significance better.

-          Line 98-line 129:

This part of the paper focuses on the role of dietary precursors in TMAO production. It does not fully consider other factors that may influence TMAO levels, such as the composition of the gut microbiome or individual differences in TMAO metabolism. It would be helpful to consider these factors better to understand the complex relationship between diet and TMAO production. This part should also discuss the potential health consequences of consuming these dietary precursors or the mechanisms by which TMAO may affect health. 

Author Response

We thank the reviewer for her/his comments and suggestions, which have helped us to improve the manuscript. Our responses are presented below, and the changes have been highlighted in the revised version of the manuscript.

 Here are some comments and suggestions for the authors:

-          Line 37: "Introduction" should be "1. Introduction"

Agree. We have corrected the mistake in the manuscript

-          Line 37-line 71:

The first comment about this part is that it presents a somewhat one-sided view of the evidence linking TMAO to ASCVD, as it only mentions studies that have found a positive association between the two. It would be helpful also to consider studies that have not found a significant association between TMAO and ASCVD to provide a more balanced view of the evidence. Additionally, this paper does not discuss potential confounding factors that may influence the relationship between TMAO and ASCVD, such as other lifestyle or dietary factors that could affect TMAO levels and ASCVD risk. It would be helpful to consider these factors to understand better the nature of the relationship between TMAO and ASCVD.

 We really appreciate the reviewer comment about this topic and we have now clarified that not all the studies have found a positive association between TMAO and ASCVD in the introduction. In the section 3 (TMAO in prospective studies) we have discussed the results from both positive and negative studies related with ASCVD. We did not discuss the influence of lifestyle or dietary factors because in the majority of the prospective studies these parameters are not well controlled and they are a common limitation in all the studies. We have included this important limitation in the section 3.

-          Line 73-line 94:

The authors should provide more context and explanations for some of the technical terms and processes mentioned in the article. Additionally, the paper does not discuss the potential health consequences of differences in TMAO production or the mechanisms by which TMAO may affect health. It would be helpful to consider this information to understand the findings' significance better.

We have simplified the description of the microbial pathways in order to be more understandable. We focused the review on ASCVD since the majority of the available prospective studies are related to cardiovascular outcomes or mortality in cardiovascular related diseases. In the section 5 we included different mechanisms by which TMAO could increase the ASCVD including cholesterol metabolism, inflammation, and thrombosis.

-          Line 98-line 129:

This part of the paper focuses on the role of dietary precursors in TMAO production. It does not fully consider other factors that may influence TMAO levels, such as the composition of the gut microbiome or individual differences in TMAO metabolism. It would be helpful to consider these factors better to understand the complex relationship between diet and TMAO production. This part should also discuss the potential health consequences of consuming these dietary precursors or the mechanisms by which TMAO may affect health.

The production of TMAO is clearly influenced by the gut composition being Firmicutes the most relevant producers of TMAO. Conversely, Bacteroidetes are associated with a low production of TMAO. Moreover, the diversity of the gut microbiota has a key role in TMAO production since a low diversity is associated with a high TMAO production. There are some studies focused on studying the impact of different types of diets in TMAO production. In the present review this topic was not acknowledged because the control of dietary habits was the most important limitation in the majority of the available prospective studies.

Reviewer 2 Report

The review is dedicated to the actual topic. It has been suggested that trimethylamine N-oxide (TMAO) promotes atherosclerosis development. Many studies have identified TMAO as a new risk factor of cardiovascular diseases. However, the underlying mechanism is still unclear.

The review is well structured material by parts on TMAO production and metabolism, prospective studies and mechanisms related to TMAO atherogenic effect. English language and style are fine, understandable for non-English-speaking readers. The article includes two large informative tables, two figures, accompanied by abbreviations. It is recommended to provide a list of the most frequently used abbreviations in a separate table at the end of the text.

Minor remarks are subject to correction.

1.    The following abbreviations need to be deciphered:

1.1.                GRP94, HSP70 – line 273;

1.2.                IL-6, CRP, TNF-α – line 316;

1.3.                NF-kB – line 318;

1.4.                ICAM-1 – line 379.

2.    ROS transcribed twice – line 314 and 333.

3.    Need to be abbreviated: tight junction pathway (TJP) – line 345,346; the abbreviation ZO-1 need to be deciphered “Zona occludens”.

Author Response

We acknowledge the reviewer for his/her remarks which have helped us to correct the manuscript. All changes have been highlighted in the revised version of the manuscript.

The review is dedicated to the actual topic. It has been suggested that trimethylamine N-oxide (TMAO) promotes atherosclerosis development. Many studies have identified TMAO as a new risk factor of cardiovascular diseases. However, the underlying mechanism is still unclear.

The review is well structured material by parts on TMAO production and metabolism, prospective studies and mechanisms related to TMAO atherogenic effect. The review is written in English, understandable for non-English-speaking readers. The article includes two large informative tables, two figures, accompanied by abbreviations. It is recommended to provide a list of the most frequently used abbreviations in a separate table at the end of the text.

A list of all the abbreviations used in the text was included in the manuscript

Minor remarks are subject to correction.

  1. The following abbreviations need to be deciphered:

GRP94, HSP70 – line 273;

IL-6, CRP, TNF-α – line 316;

NF-kB – line 318;

ICAM-1 – line 379.

We have included the full description of the abbreviations in the manuscript.

  1. ROS transcribed twice – line 314 and 333.

We have corrected it.

  1. Need to be abbreviated: tight junction pathway (TJP) – line 345,346; the abbreviation ZO-1 need to be deciphered “Zona occludens”.

We have corrected it.

Reviewer 3 Report

This article addresses the association of trimethylamine-N-oxide (TMAO) with atherosclerotic cardiovascular disease (ASCVD). The collected information is quite rich, including TMAO can cause changes in cholesterol metabolism, inflammation, endothelial dysfunction, platelet activation... and other past articles to testify. However, through these data, the author seems to be doubted with it. The author raises doubts in this article, hoping to draw everyone's attention to this issue. Therefore, the use of TMAO as a clinical indicator seems to require more research to prove it.

Specific comments

1.  Article had mention that "Trimethylamine-N-oxide (TMAO) is the main diet-induced metabolite produced by the gut microbiota, and it is mainly eliminated through renal excretion....." and "TMAO causes renal dysfunction...". Maybe can add a discussion about the kidneys and TMAO relationship. Is it because the weakened kidney function leads to an increased incidence of cardiovascular diseases or is it because of the additive effect of increased TMAO expression that aggravates the occurrence of cardiovascular diseases?

2.  Regarding the author's concerns about this, do other research teams have the same concerns? Maybe can sort out and narrate their doubts, which can arouse greater responses from readers.

3.  In figure 1, there is no way to explain that TMAO directly affects the pathway information of endothelial cells. The author only draws that TMAO will pass through the liver and then go to the kidney. Does it cause more pro-inflammatory factors after the liver?

 4. Elevated TMAO concentrations were associated with a 62% increased risk of all-cause mortality, of which cardiovascular disease was 32%, what about other causes of death?

Author Response

We really appreciate the reviewer for her/his comments which have helped us to significantly improve the manuscript. All changes have been highlighted in the revised version of the manuscript.

This article addresses the association of trimethylamine-N-oxide (TMAO) with atherosclerotic cardiovascular disease (ASCVD). The collected information is quite rich; including TMAO can cause changes in cholesterol metabolism, inflammation, endothelial dysfunction, platelet activation... and other past articles to testify. However, through these data, the author seems to be doubted with it. The author raises doubts in this article, hoping to draw everyone's attention to this issue. Therefore, the use of TMAO as a clinical indicator seems to require more research to prove it.

Specific comments

  1. Article had mention that "Trimethylamine-N-oxide (TMAO) is the main diet-induced metabolite produced by the gut microbiota, and it is mainly eliminated through renal excretion....." and "TMAO causes renal dysfunction...". Maybe can add a discussion about the kidneys and TMAO relationship. Is it because the weakened kidney function leads to an increased incidence of cardiovascular diseases or is it because of the additive effect of increased TMAO expression that aggravates the occurrence of cardiovascular diseases?

Thank you for pointing out this important topic. In the discussion of the prospective studies (Section 5) we considered the different plausible explanations. However, with the available prospective, mechanistic and genetic studies we can’t determine if TMAO is a cause or a consequence of the kidney impairment neither if TMAO can cause an increase of CVD without the implication of renal dysfunction. 

  1. Regarding the author's concerns about this, do other research teams have the same concerns? Maybe can sort out and narrate their doubts, which can arouse greater responses from readers.

The role of TMAO in ASCVD has been extensively studied over the last decade. A large number of studies have been published with positive results, while the number of negative studies is much smaller. Negative results are more difficult to be published but are needed in order to validate a hypothesis. In the recent years the number of studies that cast into doubt the role of TMAO in ASCVD have been increasing.  

  1. In figure 1, there is no way to explain that TMAO directly affects the pathway information of endothelial cells. The author only draws that TMAO will pass through the liver and then go to the kidney. Does it cause more pro-inflammatory factors after the liver?

In figure 1 we aimed to represent only the metabolism of TMAO from the ingestion of the dietary precursors until its final excretion through the kidney. TMAO can interact with endothelial cells in the circulation during its metabolism, after oxidation in the liver and before entering into kidneys. We have added this point in the figure 1 legend and shown it in the figure 2.  

  1. Elevated TMAO concentrations were associated with a 62% increased risk of all-cause mortality, of which cardiovascular disease was 32%, what about other causes of death?

These are the results of two different meta-analyses. In one meta-analysis the outcome was all-cause mortality and there was an increased risk of 62% while in the other one the outcome was cardiovascular disease and the increased risk was 23%. It should be noted that these studies pooled data from 19 and 11 available prospective cohort studies with significant heterogeneity among them and not all reported both CV events and mortality outcomes. Taking these results into account, it seems that TMAO is more related to mortality in general than specifically cardiovascular mortality.  Also, the higher risk of having cancer may also be additive in subjects with elevated TMAO levels, but this point was not addressed in these studies. The limitations of these analyses have been included in the text.